# Explainable Machine Learning to Predict Successful Weaning of Mechanical Ventilation in Critically Ill Patients Requiring Hemodialysis

**DOI:** 10.3390/healthcare11060910

**Published:** 2023-03-21

**Authors:** Ming-Yen Lin, Yuan-Ming Chang, Chi-Chun Li, Wen-Cheng Chao

**Affiliations:** 1Department of Information Engineering and Computer Science, Feng Chia University, Taichung 407102, Taiwan; 2Department of Critical Care Medicine, Taichung Veterans General Hospital, Taichung 407219, Taiwan; 3Department of Post-Baccalaureate Medicine, College of Medicine, National Chung Hsing University, Taichung 402202, Taiwan; 4Department of Automatic Control Engineering, Feng Chia University, Taichung 407102, Taiwan; 5Big Data Center, National Chung Hsing University, Taichung 402202, Taiwan

**Keywords:** mechanical ventilation, hemodialysis, weaning, machine learning, explanation

## Abstract

Lungs and kidneys are two vital and frequently injured organs among critically ill patients. In this study, we attempt to develop a weaning prediction model for patients with both respiratory and renal failure using an explainable machine learning (XML) approach. We used the eICU collaborative research database, which contained data from 335 ICUs across the United States. Four ML models, including XGBoost, GBM, AdaBoost, and RF, were used, with weaning prediction and feature windows, both at 48 h. The model’s explanations were presented at the domain, feature, and individual levels by leveraging various techniques, including cumulative feature importance, the partial dependence plot (PDP), the Shapley additive explanations (SHAP) plot, and local explanation with the local interpretable model-agnostic explanations (LIME). We enrolled 1789 critically ill ventilated patients requiring hemodialysis, and 42.8% (765/1789) of them were weaned successfully from mechanical ventilation. The accuracies in XGBoost and GBM were better than those in the other models. The discriminative characteristics of six key features used to predict weaning were demonstrated through the application of the SHAP and PDP plots. By utilizing LIME, we were able to provide an explanation of the predicted probabilities and the associated reasoning for successful weaning on an individual level. In conclusion, we used an XML approach to establish a weaning prediction model in critically ill ventilated patients requiring hemodialysis.

## 1. Introduction 

Lung and kidney are two vital and frequently injured organs in critically ill patients, and the reciprocal impacts between respiratory failure and renal failure may lead to difficult weaning among critically ill ventilated patients requiring hemodialysis [1,2]. An essential concern in critical care is the process of weaning from mechanical ventilation. Recent studies, including our own, have developed a weaning prediction model using the machine learning (ML) approach [3,4,5]. A number of clinical studies have found that acute kidney injury, particularly kidney injury that requires hemodialysis, is a substantial determinant of difficulty weaning that leads to the prolonged use of mechanical ventilation [6,7,8,9]. The difficulty in weaning among ventilated patients requiring hemodialysis may be attributed to the hemodynamic instability resulting from complex kidney–lung interactions [10,11]. 

Currently, the comprehensive hemodynamic and ventilatory parameters can be recorded by the modern information system in the intensive care unit (ICU) [12,13], but a weaning prediction model using the aforementioned electronic medical record focused on ventilated patients requiring hemodialysis is still lacking. Notably, the interpretability of the ML model has been recognized as a crucial component for the practical landing of ML models in the medical field [14]. Our previous studies have demonstrated the use of visualized interpretation to explore the model. The approach includes categorizing feature importance according to the clinical domains in the critical care workflow, illustrating the direction and strength of impacts on the target outcome among key features, and explaining the prediction at the individual level [4,15,16]. The objective of this study is to develop an extubation prediction model for critically ill ventilated patients requiring hemodialysis, utilizing an explainable machine learning approach and a multicentral critical care database. 

## 2. Materials and Methods 

### 2.1. Source of Data

The present study was conducted using the eICU collaborative research database (eICU-CRD), which consisted of the de-identified health data of approximately 200,000 admissions to 335 ICUs, including medical, surgical, and mixed ICU, at 208 hospitals across the United States between 2014 and 2015. All tables in the aforementioned database were de-identified and meet the requirements of the Health Insurance Portability and Accountability Act (HIPAA) in the United States. Data consisted of static information (i.e., demographics, severity scores, and comorbidities), daily obtained information (i.e., laboratory and ventilatory parameters), and data obtained with a high sampling rate (i.e., hemodynamic parameters). All data were extracted by the Structured Query Language with PostgreSQL (version 13.3). 

### 2.2. Machine Learning Models 

In this study, we utilized four machine learning (ML) models, consisting of extreme gradient boosting (XGBoost), gradient boosting machine (GBM), adaptive boosting (ada-boost), and random forest (RF). In general, the four models were often used in classification and prediction tasks because they have been found to be highly effective and accurate in many real-world applications. Moreover, these models were tree-based ML models, which were more likely to be explained and deployed in the information system at the hospital. The training-to-testing data proportion was set at 80/20 (Appendix A. provides details of the parameters in the aforementioned ML models). Our aim was to predict weaning two days prior to extubation using data from three and four days prior to extubation. Therefore, both the feature window and prediction window were set at 48 h (Appendix A provides details of the study’s data time frame). To address concerns about the black box issue in ML models, we used several visualization tools to explain the ML models at the domain, feature, and individual levels. We computed the feature importance scores and illustrated the cumulative feature importance in accordance with the main clinical domains, including the ventilatory parameter, laboratory data, and hemodynamic data domains. These domains are fundamental in considering weaning in critical care. 

We chose XGBoost in the following experiments due to its high flexibility of XGBoost. To disentangle the black box issue at the feature level, we utilized PDP and SHAP plots. These plots demonstrate the strength and direction of impacts on the weaning outcome at the feature level [17]. The SHAP summary plot provided a comprehensive representation of the direction and magnitude of correlations between features and weaning outcomes, while the partial dependence plot (PDP) further illustrated the marginal effect of the feature on the weaning outcome. To provide an intuitive explanation at the individual level, we not only listed the weaning probability among individuals, but also used LIME plots to show the impact of key features on the weaning outcome [18]. In brief, LIME provides a model-agnostic explanation of the classification model by applying a locally linear model. The LIME plot illustrates the contribution of key features to the prediction of the weaning outcome for a selected individual. 

### 2.3. Statistical Analysis

Here, the continuous data are presented as means ± standard deviations, while categorical data are shown as frequencies (percentages). To evaluate the difference between the two groups, two statistical techniques were employed, specifically Fisher’s exact test and Student’s *t*-test. The evaluation of the models’ discrimination and accuracy in the testing sets was conducted using receiver operating characteristic (ROC) curve analysis and calibration curve, respectively. The Delong test was further employed to evaluate the performance differences among distinct machine learning models. We also calculated the accuracy, recall, F1 score, positive predictive value, and negative predictive value in distinct models. In addition to the ROC curve, we also plotted the calibration curve, which was a reliability diagram to demonstrate the relative frequency of observed versus the predicted probability. The Python version used in the study was 3.7.4.

## 3. Results 

### 3.1. Demographics and Basic Characteristics 

A total of 1789 critically ill patients requiring both hemodialysis and mechanical ventilation during ICU admission in the 335 ICUs across the United States were eligible for analyses (Figure 1). The enrolled subjects had a mean age of 62.1 ± 14.2 years, with 56.9% being male. Given that we excluded those requiring mechanical ventilation-alone or hemodialysis-alone, the enrolled subjects showed a high degree of disease severity, as evidenced by their acute physiology and chronic health evaluation (APACHE) IV score of 83.8 ± 37.2. 

We found that 42.8% (765/1789) were successfully weaned from mechanical ventilation during ICU admission. There were no significant differences in age and sex distributions between patients who were successfully weaned and those who were not. However, those who cannot be weaned from mechanical ventilation had a higher APACHE IV score (89.9 ± 39.3 vs. 75.6 ± 32.4, *p* < 0.001) and were less likely to have end-stage renal disease (22.4% vs. 39.3%, *p* < 0.001), compared to those with successful weaning (Table 1). With regard to ventilator parameters, patients without successful weaning received a higher fraction of inspired oxygen (FiO_2_) (52.0 ± 20.6 vs. 43.6 ± 16.0%, *p* < 0.001), positive end-expiratory pressure (PEEP) (6.5 ± 2.9 vs. 5.5 ± 1.6 cmH_2_O, *p* < 0.001), tidal volume VT (489.5 ± 124.5 vs. 472.1 ± 146.7 mL, *p* < 0.001), peak airway pressure (Ppeak) (21.2 ± 5.3 vs. 19.3 ± 4.8 cmH_2_O, *p* < 0.001), and mean airway pressure (Pmean) (11.1 ± 3.5 vs. 9.7 ± 2.4 cmH_2_O, *p* < 0.001) than those weaned from mechanical ventilation successfully. Furthermore, patients who cannot be weaned from mechanical ventilation were more likely to have abnormal laboratory data, lower blood pressure, and a higher heart rate than those with successful weaning. 

### 3.2. Performance of the Four Machine Learning Models 

We conducted a comparative analysis of the performance of the four machine learning (ML) models to forecast successful extubation. Notably, the AdaBoost/random forest models demonstrated low accuracy, while the XGBoost and GBM models displayed relatively similar high accuracy, with AUC of 0.83 and 0.81, respectively (as illustrated in Figure 2A). Comprehensive performance metrics are detailed in the Appendix A, while the outcome of the Delong test is presented in Appendix A. Furthermore, the calibration curve demonstrated a satisfactory agreement between the predicted values and observed values in both XGBoost and GBM (as indicated in Figure 2B).

### 3.3. Global Explanation of the ML Model at the Clinical Domain and Feature Level 

The ML model was then examined at various levels, including the clinical domain level, feature level, and individual level. The top 20 features were categorized by main domains, based on the clinical workflow for weaning in critically ill ventilated patients. We found that the proportion of cumulative feature importance of the ventilation domain, laboratory domain, and vital signs domain were 36.8%, 33.7%, and 10.7%, respectively (Figure 3). We utilized the SHAP summary plot to demonstrate the impact of the key features on the probability of weaning (Figure 4). The SHAP plot not only displayed the direction, but also the strength of each feature’s influence on the weaning outcome. For example, a lower FiO_2_, Pmean, level of total bilirubin, and ventilator day were associated with a higher probability of successful weaning. We further used PDP plots of the six high-ranking features to provide a more detailed understanding of how each feature impacts the probability of successful extubation (Figure 5). 

For example, the patients with FiO_2_ lower than approximately 40–50% or Pmean lower than 10–15 cmH_2_O tended to be successfully weaned from mechanical ventilation. Taken together, the aforementioned visualized explanations of the clinical workflow for weaning in critical care at both the domain and feature levels should provide clinicians with an intuitive interpretation of the ML model.

### 3.4. Local Explanation of the ML Model for Successful Extubation Prediction of the Individual Patient 

To demonstrate the overall influence of key features on the weaning prediction model, we utilized LIME plots for two representative patients. The overall predicted probability of extubation, as well as the incremental effects (green) on successful weaning of variables and decremental effects (red) on extubation of variables for the two patients, were shown in Figure 6. For example, in patient-187, the predicted probability for extubation was relatively low (0.27), due to several unfavourable conditions, involving a high APACHE IV score (113), long ventilator day (8 days), and receiving sedation, although a relatively low Pmean (8 cmH_2_O) and PEEP (5 cmH_2_O) (Figure 6A). In contrast, the probability of successful weaning in patient-230 was relatively high (0.73), due to a number of favourable conditions, including low PEEP (5 cmH_2_O), low APACHE IV score (43), low white blood cell count (6900 per mL), and low heart rate (71 per minute), despite the need for sedation (Figure 6B). The collective explanations offered at the individual level are consistent with both the feature-level explanations and the clinical workflow in critical care. As such, these insights can help to significantly mitigate the black box issue associated with this model.

## 4. Discussion 

To predict successful weaning is an essential issue in critically ventilated patients requiring hemodialysis. We established an explainable ML-based weaning prediction model with a 48-h prediction window using the database of 335 ICUs across the United States. To increase the transparency of the model and facilitate the practical landing in the near future, we illustrated the importance of feature importance based on clinical working domains in critical care and employed SHAP, as well as PDP, plots for the visualized interpretability of key features and used LIME for the explanation in individual subjects. This evidence provides the application of explainable ML for weaning prediction in critically ventilated patients requiring hemodialysis, and the interpretability along with clinical workflow in critical care should mitigate the black box issue in the practical landing of medical AI. 

The lung and kidney are two frequently injured vital organs in critical illness, and the concomitant dysfunction of these two organs is associated with significant morbidity and mortality in critically ill patients [2,19]. A number of evidence have shown that patients with respiratory failure have an approximately threefold increase in the odds of developing acute kidney injury (OR 3.16, 95% CI 2.32–4.28), and a number of pathophysiological mechanisms, including haemodynamic, pulmonary edematous, inflammatory, immunological, and neurohormonal effects, have been implicated with the kidney–lung interaction [1,20]. Accumulating studies have identified the deleterious impact of hemodialysis on the weaning outcome in critically ill ventilated patients, including our previous study focusing on patients requiring prolonged mechanical ventilation (PMV); therefore, there is a research niche to establish a weaning outcome prediction model focusing on critically ill ventilated patients requiring hemodialysis [4,6,8]. Given that patients with renal failure and respiratory failure may be distinct from those with respiratory failure alone, the weaning model should be established specifically in critically ventilated patients requiring hemodialysis. For example, hemodialysis is generally applied approximately every two days; therefore, we used 48-h as the feature window prediction window in establishing the weaning outcome prediction model in critically ill ventilated patients requiring hemodialysis. 

Intriguingly, we found that low variability, instead of the average level, of oxygen saturation and systolic blood pressure are crucial predictors for successful weaning from mechanical ventilation in critically ill ventilated patients. This finding is in line with previous studies, including our study, showing that glycemic variability was associated with a poor outcome in critically ill patients with sepsis and coronavirus disease (COVID) [21,22]. Similarly, Park S. et al. recently reported that heart rate variability was associated with incident intradialytic hypotension among 71 patients with renal failure [23]. Indeed, the intensivist often attempts to keep the vital signs, including oxygen saturation, blood pressure, and heart rate, within normal range by distinct medications and management. Therefore, the average level of vital parameters tended to be within the normal range, and the variability may be more likely to reflect the underlying critical illness. 

The finding mentioned above in the present study highlights the practical application of stream data. The high-frequency stream data, with respect to hemodynamic parameters in the e-ICU database, enable us to include dynamic parameters in the weaning prediction model [24]. Notably, modern ICUs have stream data, with regards to hemodynamic parameters; however, the majority of studies applying AI models in critical care mainly used static, instead of dynamic, data to establish the prediction model, given that stream data might potentially lead to difficulty in data processing [24]. In line with the 90-day mortality prediction model proposed by Thorsen-Meyer et al. [24], we demonstrated the feasibility of using a surrogate dynamic parameter, such as the variability of oxygen and blood pressure, of stream data within a certain time period to establish the model, and this approach should mitigate the concern of real-world application of stream data. Indeed, a similar approach has been employed in critical care. For example, the variation of pulse pressure (PPV) and variation of stroke volume (SVV) have been widely applicated as dynamic parameters for fluid resuscitation in patients with sepsis [25,26,27].

To mitigate the black box issue and to facilitate human–artificial intelligence (AI) teamwork, the interpretation of the ML model is currently an essential issue in the application of ML in the medical field [14,28]. In the present study, we presented the feature importance of clinical domains, including ventilation parameters, laboratory data, and vital signs, and these domains are exactly in accordance with the clinical workflow of decision-making, with regard to weaning by the intensivist (Figure 3). In addition to the interpretation at the domain level, we further specified the PDP that allows the physician to realize the cut-point of the six top-ranking features (Figure 4). For example, we found that total bilirubin higher than 2 mg/dL, which is exactly the criteria to define gastrointestinal dysfunction in SOFA score, was associated with a decrease of weaning probability (Figure 5C) [29]. Similarly, the count of white blood cells higher than 15,000 may indicate the underlying infectious or inflammatory conditions and is associated with a reduced probability of weaning from mechanical ventilation (Figure 5E). Collectively, we think the visualized interpretation at the feature level by PDP should further mitigate the black box issue and facilitate the feasibility of human–AI teamwork. 

The black box issue impedes practical implementation in healthcare applications; therefore, the human-understandable explanation is currently a growing research area to provide an explainable, trustable, and traceable intelligent model in medical fields, including the prediction of coronavirus disease (COVID) infection [30,31]. We used the real-world dataset in this study to establish an explainable ML-based weaning prediction model in critically ill ventilated patients requiring hemodialysis. Similarly, Rotami M. et al. recently used real-world blood test data and an explainable ML-based approach to predict patients with a COVID diagnosis [30]. In detail, they exploited graph analysis for feature visualization and employed an explainable decision forest classifier to predict patients with COVID [30]. As shown in the aforementioned study and the present study, the human-understandable explanation consists of not only global explanations aiming to explain the whole model, but also local explanations focusing on a small area of the individual sample. We think the global explanation may enhance the trust in the ML model, and the local explanation may incorporate the ML model into the workflow of healthcare.

The goal of an AI-based decision support system is to provide individualized suggestions for patients with complex critical illnesses, such as critically ill ventilated patients requiring hemodialysis in the present study [31]. Nevertheless, the design of the decision support system should be in line with the workflow in critical care settings for the practical application of the mode. For example, the prediction window is two days in the present study, and we think the adequate prediction window enables the physician to start the spontaneous breathing trial, as well as to make the final decision regarding weaning (Appendix A) [32]. Therefore, the proposed ML-based prediction should facilitate the weaning from the mechanical ventilator in critically ill ventilated patients requiring hemodialysis. Additionally, the established model used the real-world data collected during the daily practice at the 335 ICUs, and the interoperability for implementation should be less likely to be a concern. Another issue for the practical application, also called landing, is the user interface in real-world conditions [31]. Given that the focus of this study is the interpretability of the ML-based model, both the global and local explanations can be illustrated. We think the understandable model, through enhancing model explainability, should have a high possibility of landing in the future. 

There are limitations that merit discussion. First, the number of enrolled subjects was relatively small. Due to merely 5–10% of critically ill ventilated patients requiring hemodialysis, the number of participants in studies focusing on patients with concomitant dysfunction of these two organs tends to be small. Given that the proportion of patients requiring mechanical ventilation and hemodialysis in critically ill patients in this study was similar to the data in previous studies, we think the concern of generalization of the findings in the present study should be at least partly mitigated. Second, the retrospective nature of this study and further prospective studies are warranted to validate our findings. Third, some hemodialysis-relevant data, such as intradialytic hypotension, were not included in the dataset. Fourth, the technology readiness level (TRL) of the weaning prediction ML model among critically ill ventilated patients requiring hemodialysis should be TRL-4 with model development [33], and more efforts, including external validation and establishment of an interface for real-time model testing, are warranted. However, we think the feasibility of the practical application of our findings might be high, due to the fact that the variables used for model development in this study were retrieved from the structured and real-world electronic medical records in the ICUs across the United States. Fifth, the proposed model could be further clarified by the employment of the ablation experiment [34]. However, the ablation approach might be at least partly unintuitive for the intensivist, given that critical care relevant variables are deemed critical parameters; hence, we chose to provide the cumulative feature importance of main clinical domains in critical care in the present study (Figure 3).

## 5. Conclusions

In conclusion, we used a multi-center dataset and explainable ML approach to establish a weaning prediction model in critically ill ventilated patients requiring hemodialysis. We found that XGBoost outperformed the other ML models for predicting successful weaning. By visualizing the features that contributed to individual-specific predictions, in accordance with the clinical workflow of critical care, we believe our explainable approach can help mitigate the black box issue and promote the adoption of our proposed model in the near future. Future prospective research is warranted to translate the explainable ML models into a practical decision support system in critical care.

## Figures and Tables

**Figure 1 healthcare-11-00910-f001:**
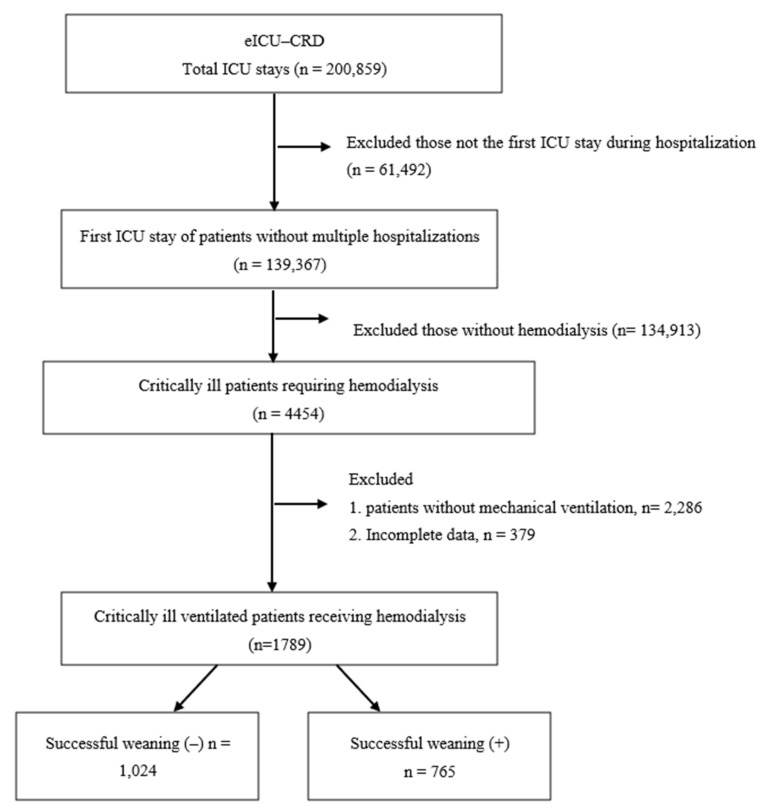
Flowchart showing the selection of critically ill ventilated patients receiving hemodialysis from the eICU Collaborative Research Database. Patients were excluded if they had multiple hospitalizations or if the current ICU stay was not their first. The number of patients who were successfully weaned from mechanical ventilation is shown. Abbreviations: eICU-CRD, eICU Collaborative Research Database; ICU, intensive care unit; (+), successful weaning from mechanical ventilation.

**Figure 2 healthcare-11-00910-f002:**
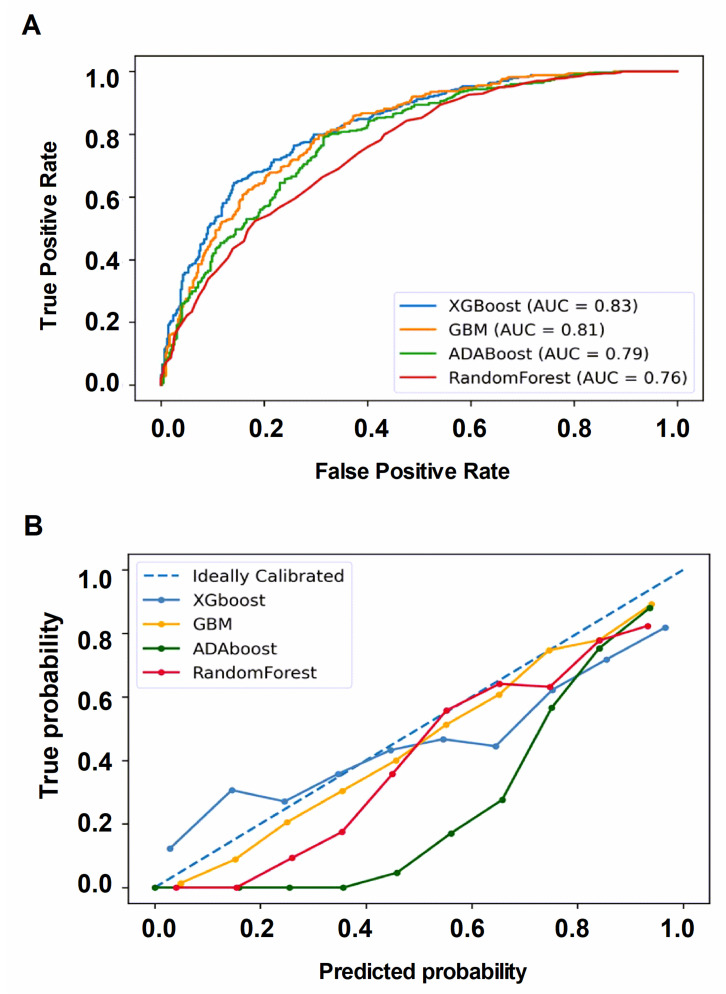
(**A**). The receiver operating characteristic curve was used to illustrate the performance of classifiers at various threshold settings, and AUC represents the degree of performance (**B**). The calibration curve was a reliability diagram to show the relative frequency of observed versus the predicted probability. Abbreviations: XGBoost, eXtreme Gradient Boosting; GBM, Gradient Boosting Machine; AdaBoost, Adaptive Boosting; RF, Random Forest.

**Figure 3 healthcare-11-00910-f003:**
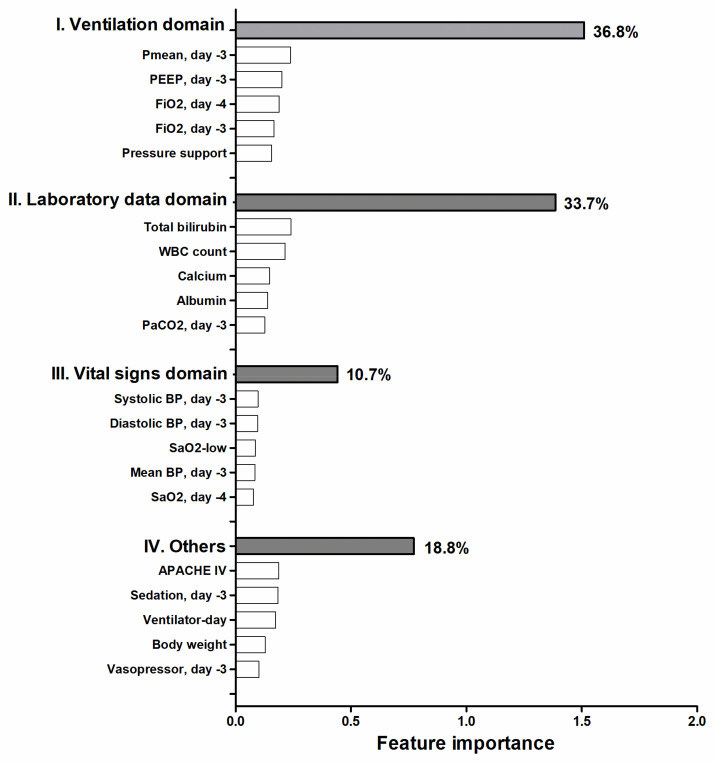
Cumulative relative feature importance of top 20 features categorized by main clinical domains in predicting successful weaning. Feature importance of each feature indicated how much each feature contributed to the model prediction.

**Figure 4 healthcare-11-00910-f004:**
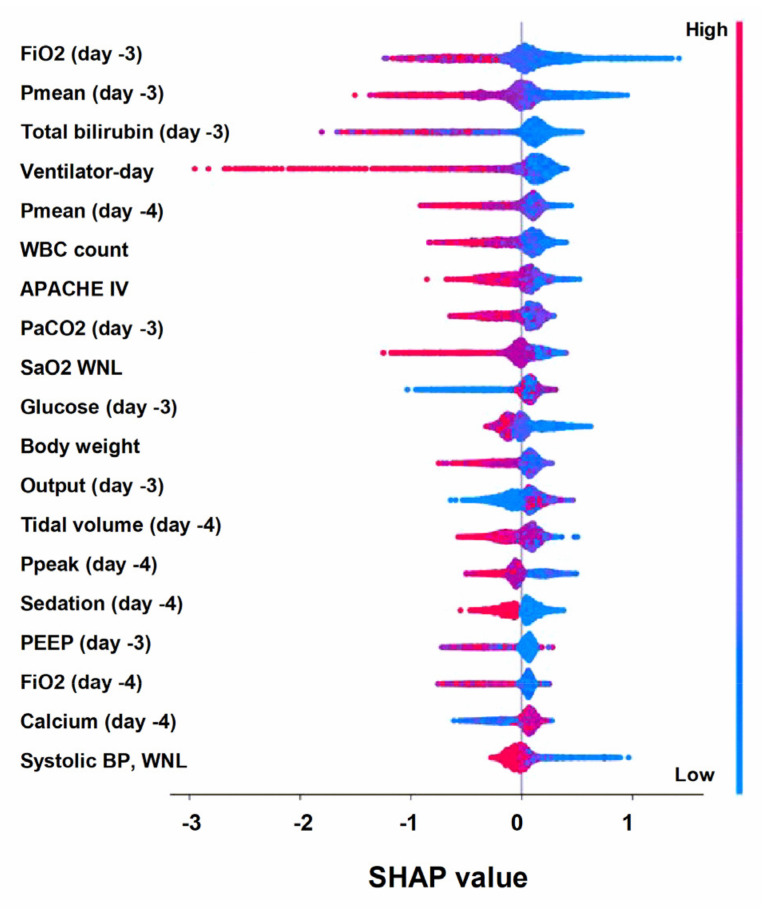
SHAP to illustrate the one–year mortality prediction model at the feature level. Each point on the plot represented a Shapley value for one feature and subject. Abbreviation: SHapley Additive exPlanation (SHAP).

**Figure 5 healthcare-11-00910-f005:**
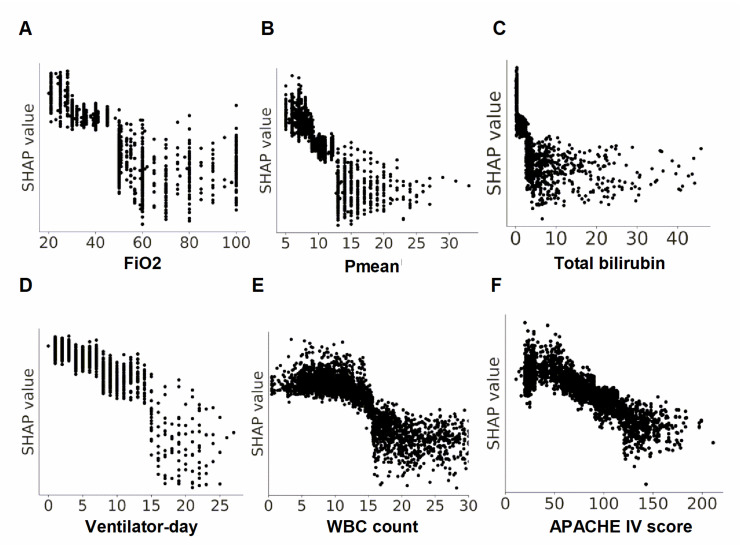
Partial dependence plots of key features (**A**). FiO_2_, (**B**). Pmean, (**C**). Total bilirubin, (**D**). Ventilator day, (**E**). WBC count. (**F**). APACHE IV score. The partial dependence plot showed the marginal effect of each feature on the weaning outcome. Abbreviation: FiO_2_, the fraction of inspired oxygen; Pmean, mean airway pressure; WBC, white blood cell; APACHE IV, acute physiology, and chronic health evaluation IV.

**Figure 6 healthcare-11-00910-f006:**
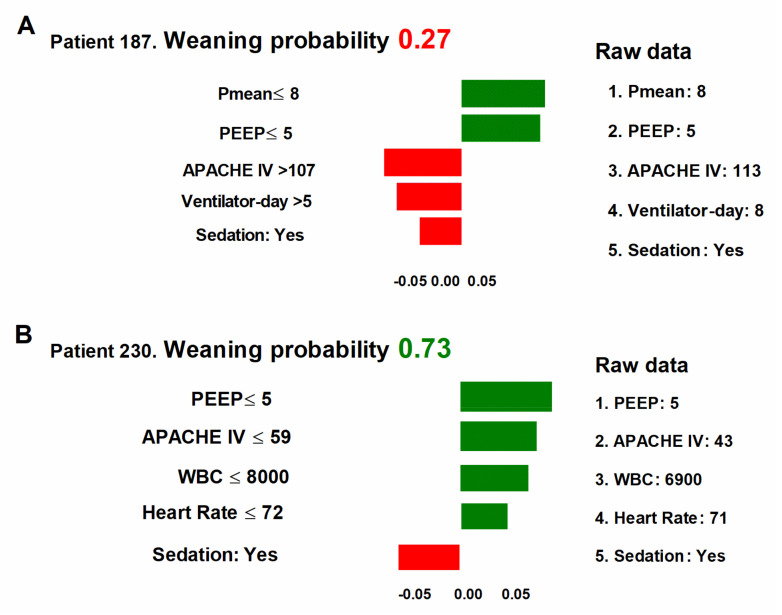
Local interpretable model–agnostic explanations (LIME) of two representative individuals (**A**) patient 187; (**B**) patient 230. LIME was used to visualize the individual predictions among the top 5 features. The overall predicted probability of extubation was listed on the top, followed by the variables with incremental effects (green) or decrement effect (red) on successful weaning.

**Table 1 healthcare-11-00910-t001:** Characteristics of the 1789 critically ill ventilated patients receiving hemodialysis, categorized by weaning outcome.

	All	Successful Weaning (−)	Successful Weaning (+)	*p*-Value
	N = 1789	N = 1024	N = 765	
Demographic data				
Age (years)	62.1 ± 14.2	62.3 ± 14.1	61.8 ± 14.3	0.393
Sex (male)	1018 (56.9%)	568 (55.5%)	450 (58.8%)	0.156
Body weight (kg)	88.2 ± 27.7	90.4 ± 29.8	85.4 ± 24.3	<0.001
Height (cm)	169.6 ± 11.3	169.8 ± 11.8	169.4 ± 10.6	0.453
End-stage renal disease	530 (29.6%)	229 (22.4%)	301 (39.3%)	<0.001
APACHE IV	83.8 ± 37.2	89.9 ± 39.3	75.6 ± 32.4	<0.001
Ethnicity				0.520
African American	254 (14.2%)	144 (14.1%)	110 (14.4)	
Asian	17 (1.0%)	8 (0.8%)	9 (1.2%)	
Caucasian	1094 (61.2%)	644 (62.8%)	450 (58.8%)	
Hispanic	79 (4.4%)	39 (3.8%)	40 (5.2%)	
Native American	28 (1.6%)	22 (2.1%)	6 (0.8%)	
Other	317 (17.8%)	167 (16.3%)	150 (19.6%)	
Ventilatory parameters				
FiO_2_ (%)	48.4 ± 19.2	52.0 ± 20.6	43.6 ± 16.0	<0.001
PEEP (cmH_2_O)	6.1 ± 2.5	6.5 ± 2.9	5.5 ± 1.6	<0.001
VT (mL)	482.1 ± 134.7	489.5 ± 124.5	472.1 ± 146.7	0.007
Ppeak, (cmH_2_O)	20.4 ± 5.2	21.2 ± 5.3	19.3 ± 4.8	<0.001
Pmean, (cmH_2_O)	10.5 ± 3.2	11.1 ± 3.5	9.7 ± 2.4	<0.001
Laboratory data				
White blood cell count (count/μL)	15.9 ± 10.2	17.3 ± 11.2	14.0 ± 8.3	<0.001
Hemoglobin (g/dL)	9.9 ± 2.0	10.0 ± 2.0	9.9 ± 1.9	0.587
Platelet (103/μL)	181.8 ± 111.8	172.6 ± 111.2	194.1 ± 111.5	<0.001
Total bilirubin (mg/dL)	2.3 ± 4.5	2.9 ± 5.2	1.6 ± 3.4	<0.001
Lactate (mmol/L)	4.4 ± 4.1	5.1 ± 4.8	3.4 ± 2.7	<0.001
HCO_3_ (mmol/L)	23.7 ± 4.7	23.1 ± 5.0	24.3 ± 4.2	<0.001
PaCO_2_ (cmH_2_O)	44.7 ± 13.5	45.1 ± 14.3	44.2 ± 12.4	0.188
Vital signs and fluid balance				
Systolic blood pressure (mmHg)	118.5 ± 28.1	112.8 ± 26.3	126.1 ± 28.6	<0.001
Diastolic blood pressure (mmHg)	61.9 ± 18.3	59.4 ± 16.7	65.3 ± 19.7	<0.001
Mean blood pressure (mmHg)	78.1 ± 22.1	74.1 ± 19.4	83.5 ± 24.2	<0.001
Heart rate (per minute)	90.8 ± 20.8	93.6 ± 21.2	87.1 ± 19.6	<0.001
Respiratory rate (per minute)	17.8 ± 6.3	17.8 ± 6.5	17.9 ± 5.9	0.826
Oxygen saturation (%)	96.6 ± 6.1	96.1 ± 6.5	97.3 ± 5.6	<0.001
Daily input, mL	1938.6 ± 1729.4	2112.7 ± 1832.7	1705.4 ± 1550.2	<0.001
Daily output, mL	1096.6 ± 1677.2	1147.1 ± 1806.6	1029.0 ± 1484.0	0.141
Ultrafiltration, mL	−2090.6 ± 1025.4	−2058.4 ± 975.8	−2133.8 ± 1086.8	0.51
Outcome				
ICU length of stay (day)	7.8 ± 7.9	9.5 ± 8.6	5.6 ± 6.2	0.118
Ventilator day (day)	6.0 ± 6.2	7.9 ± 7.3	3.4 ± 2.8	0.004
Hospital stay (day)	17.3 ± 16.7	17.8 ± 17.1	16.5 ± 16.1	0.747

Data were presented as mean ± standard deviation and number (percentage). Abbreviations: APACHE IV, acute physiology, and chronic health evaluation IV; FiO_2_, the fraction of inspired oxygen; PEEP: positive end-expiratory pressure; V_T_: tidal volume; P_peak_: peak airway pressure; Pmean, mean airway pressure; HCO_3_; Bicarbonate; PaCO_2_, partial pressure of carbon dioxide; ICU, intensive care unit.

## Data Availability

All of the data and materials are provided in the manuscript and the Appendix A. The code has been put in public Github and is available via https://github.com/GitEricLin/Predict_Weaning_HD (accessed on 18 March 2023).

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
