# Peer review of "Explainable Machine Learning to Predict Successful Weaning of Mechanical Ventilation in Critically Ill Patients Requiring Hemodialysis"

_healthcare, 2023, doi:10.3390/healthcare11060910_

Round 1

Reviewer 1 Report

The main contribution of the paper consists in establishing a weaning prediction model for critically ill ventilated ICU patients on hemodialysis. The proposed prediction model is based on an explainable machine learning approach (XML), which has been employed in the current research and other recent studies focusing on weaning from ventilation. Compared to these other studies, the proposed research used medical recordings taken with current ICU information systems of hemodynamics and ventilation parameters. Considering that there is a lack of studies that use such medical recordings in XML weaning prediction models, and the paper brings a needed contribution in this area.
    The more specific contributions of the research consist in model explanations addressing domain, feature, and individual levels. The authors have been able to synthesize the discriminative points for several key features in weaning prediction. At individual level, the authors provide predictions of probabilities and prediction rationales for successful weaning. Based on the research presented in this paper, the authors envision the future development of a support system to ground decision making in ICU care.
    The data sources for the research are clearly explained. The research is based on the publicly available eICU collaborative research database and the authors provide sufficient information to estimate the relevance of the data and how it is used for the purposes of the proposed research. In this sense, the authors correctly acknowledge that the number of patients on ventilation, who also require hemodialysis, is small compared to overall data sample size and tends to be small for the purposes of their study. 
    The research platform and the employed research methodology are also explained with clarity. As much as possible, the authors used data visualization methods for a more intuitive comprehension of the meaning of XML model' explanations and this is a good aspect of the paper. The authors provide sufficient information for the research to be reconstructed and tested, but the reviewer did not undertake this task.   
    The abstract provides a good synthesis of the research presented in the paper. The conclusion section provides an effective summary of the research results and possible future research development. References are current, with most of them being published during the last five years.

Reviewer 2 Report

The authors provided explanations of the model at the domain, feature, and individual levels by using cumulative feature importance, Shapley additive explanations (SHAP) plot, partial dependence plot (PDP) as well as local interpretable model-agnostic explanations (LIME).  The concept is good but still needs some editing and improvements:

1. The authors computed the True-Positive rate to validate the developed method. The authors should clearly mention why this measure was used to evaluate. Why not F1, Precision, Recall, ROC, or MCC? If it is relevant, authors should consider different evaluation measures to compare their model with other existing models and should tabulate.
2- Explain how each of the parameters influences the performance of the proposed approach.

3- Provide an ablation study to investigate the impact of the different parts of your model.

4- In connection with the review of related papers on this manuscript, several examples are suggested that the author is better to add.

--A novel explainable COVID-19 diagnosis method by integration of feature selection with random forest. 2022,  Informatics in Medicine Unlocked, 30, p.100941.

-- An effective explainable food recommendation using deep image clustering and community detection, 2022, Intelligent Systems with Applications, 16, p.200157.

5- The discussion section should include some future works.

Reviewer 3 Report

Dear Editor,

The manuscript " Explainable machine learning to predict successful weaning of mechanical ventilation in critically ill patients requiring hemodialysis" by  Lin MY et al. analyzes patients who in intensive care required dialysis and artificial ventilation due to respiratory and renal insufficiency.

The paper analyzed hemodynamic parameters, biochemical and elements from ventilation support. The success of the recovery of respiratory function and extubation in these patients was investigated through a multicenter database that was analyzed using ML models for prediction. The cumulative effect and reliability of the tested connectivity parameters, which are important for assessing the reliability of extubation and the outcome, were proven. The characteristics of patients who needed prolonged respiratory support are given. Different machine learning models verified the obtained results in the discussion, the obtained results were compared with the results of studies published so far, and the shortcomings of the study were mentioned.

I recommend explaining what kind of patients are treated in intensive care: surgical or internist and why the mentioned machine learning models were used.

I would like to thank the editor for the opportunity to review the work and the authors for their efforts in this excellent work. 

Reviewer 4 Report

General Comments 

Reviewed is the revised manuscript “Explainable machine learning to predict successful weaning of mechanical ventilation in critically ill patients requiring hemodialysis” submitted by Ming-Yen Lin, et. al. The article created a machine-learning weaning prediction model for critically sick patients with respiratory and renal failure. The study included 1,789 patients and included a variety of explanation strategies. The authors concluded that the XML technique could be utilized to create such a model. The paper is well-structured, with a smooth flow of information. Although some grammatical issues need to be fixed before publication, the methodology and findings presented in the paper are significant to the field of ranking and selection. The authors have effectively presented their findings, making the manuscript a strong contender for publication with minor revisions.

Minor revisions:

·         Some texts in Figure 1 are cropped, making the text below "Critically ill patients requiring hemodialysis (n= 4,454)" not visible.

·         In Figure 1 legend, the legend does not describe what the figure is showing, such as the purpose or main findings of the study.

A revised legend could be something like this "Figure 1. Flowchart showing the selection of critically ill ventilated patients receiving hemodialysis from the eICU Collaborative Research Database. Patients were excluded if they had multiple hospitalizations or if the current ICU stay was not their first. The number of patients who were successfully weaned from mechanical ventilation is shown. Abbreviations: eICU-CRD, eICU Collaborative Research Database; ICU, intensive care unit; (+), successful weaning from mechanical ventilation.

·         The legends for figures 3-6 have comparable issues.

·         The limitations section could be expanded to include other potential areas of future research, which would provide a more complete picture of the study's implications.
